# Do Sleep Disturbances Improve Following Psychoanalytic Psychotherapy for Adolescent Depression?

**DOI:** 10.3390/ijerph19031790

**Published:** 2022-02-04

**Authors:** Thea Schønning, Hanne-Sofie Johnsen Dahl, Benjamin Hummelen, Randi Ulberg

**Affiliations:** 1Institute of Clinical Medicine, University of Oslo, P.O. Box 1171, 0318 Oslo, Norway; randi.ulberg@medisin.uio.no; 2Department of Psychology, University of Oslo, Forskningsveien 3, 0370 Oslo, Norway; h.s.j.dahl@psykologi.uio.no; 3 Vestfold Hospital Trust，Research Unit, Division of Mental Health and Addiction, Vestfold Hospital Trust, P.O. Box 2169, 3125 Tønsberg, Norway; 4Department of Psychiatry, Diakonhjemmet Hospital, Forskningsveien 7, 0370 Oslo, Norway; 5Department of Research and Innovation, Division of Mental Health and Addiction, Oslo University Hospital, P.O. Box 4959, 0424 Oslo, Norway; UXBEUM@ous-hf.no

**Keywords:** adolescence, depression, psychotherapy, transference, sleep

## Abstract

Sleep disturbance is often a prominent symptom in adolescents diagnosed with major depressive disorder (MDD). Recent evidence indicates that short-term psychoanalytic psychotherapy (STPP) for depression may have an effect in reducing co-occurring sleep disturbance in youth. It is unknown if transference work (exploration of the patient–therapist relationship) has an additional effect in reducing sleep disturbance. Adolescents aged 16–18 years (*n* = 69, 84% female) who met diagnostic criteria for MDD based on the Mini International Neuropsychiatric Interview (M.I.N.I) were randomized to either STPP with transference work or without. Sleep problems were assessed at baseline, therapy session 20 (20 weeks), post-treatment (28 weeks), and one-year follow-up (80 weeks) with the Symptom Checklist-90-R. At baseline, 69% of the adolescents exhibited moderately to extreme sleep difficulties. Sleep disturbance was significantly correlated to depression depth at session 20 and at follow-up. Symptoms of insomnia significantly decreased from baseline to the end of treatment. Treatment gains were maintained until follow-up. No differences in recovery of sleep disturbance were found between the two treatment groups. The findings suggest that sleep disturbance improves following STPP for depression, with or without transference work. Future research should assess those with residual symptoms by different sleep measures.

## 1. Introduction

Sleep disturbance is one of the most common symptoms in adolescents diagnosed with a depressive illness [1,2]. To diagnose major depressive disorder (MDD), the adolescent must present with at least one core depressive symptom alongside with additional symptoms, such as sleep disturbance [3,4]. Sleep disturbances refer to difficulties such as too little sleep or fragmented sleep (insomnia), too much sleep (hypersomnia), not feeling rested after sleep (non-restorative sleep), and shifts in sleep timing (circadian reversal) [2,5,6]. Insomnia is the most common and by far the most frequently investigated depression-specific sleep symptom in the literature. Insomnia includes difficulty falling asleep (initial insomnia), difficulty maintaining sleep (middle insomnia), and waking up too early (terminal insomnia). Although recognizing the impact of other relatively common co-occurring sleep disturbances, this paper will principally consider the three types of insomnia [7,8]. 

There is a growing body of literature that recognizes the importance of sleep disturbance and how it is related to adolescent depression. Several studies with a longitudinal design argue that impaired sleep affects the initiation, development, and maintenance of depression in adolescents [8,9,10,11]. In their meta-analysis, Lovato and Gradisar proposed a conceptual model to understand how lack of sleep leads to rumination, negative affect, hyperarousal, and emotional dysregulation, with subsequent depression and insomnia as a result [9]. However, recent evidence for the adult population strongly indicates a bidirectional relationship between sleep disturbance and MDD in which rumination in turn leads to sleep disturbance [12].

Young individuals experiencing disturbed sleep are more likely to suffer from severe depression [10]. Compared with adolescents without sleeping problems, individuals experiencing sleep disturbance are also more likely to have thoughts about suicide [13], illustrating the importance of focusing on sleep problems in adolescents. There is a need to strengthen our knowledge of how sleep disturbance is related to symptoms of depression during treatment, and what treatment works for whom. 

The Improving Mood with Psychoanalytic And Cognitive Therapies (IMPACT) was the first study that explored how psychotherapy for depression may reduce sleep problems in young individuals [1]. Participating adolescents diagnosed with moderate to severe MDD received one of three treatment approaches: cognitive behavioral therapy (CBT), short-term psychoanalytic psychotherapy (STPP), or the active control treatment brief psychosocial intervention (BPI). The IMPACT study found that the number of sleep problems in adolescents significantly reduced during treatment for depression, measured by semi-structured interviews and self-report forms. Approximately half of the participating adolescents reported a reduction in sleep disturbances after receiving treatment for depression. However, half of the participants reported residual symptoms of sleep disturbance after treatment, and one-third of adolescents experienced sleep problems after recovering from MDD. The IMPACT study found no significant main effect between CBT, STPP, or the active control treatment BPI, equivalent to the effectiveness of the three treatments found for depression severity [14]. To the best of our knowledge, no previous studies have examined if the outcomes seen in the two IMPACT trials were due to specific psychotherapeutic techniques, or if common factors in psychotherapy led to improvement in both depressive symptoms and sleep disturbance. 

Transference work is a specific technique in psychoanalytic psychotherapy that is assumed to facilitate change. Transference work involves the active focus on the patient–therapist relationship in-session. When young patients are invited to do so, exploration of the therapeutic relationship may comprise a balance between autonomy and acknowledgment, as well increased self-acceptance. This is thought to be of importance for relieving depressive symptoms [15]. 

The First Experimental Study of Transference work—In Teenagers (FEST-IT) recently assessed the effects of transference work in STPP for adolescents diagnosed with MDD [15]. The FEST-IT study is a randomized clinical trial (RCT) in which 69 adolescents between 16 and 18 years were randomized to either STPP with or without transference interventions. In a recent publication from the FEST-IT, it was shown that depressive symptoms reduced in both treatment groups [15]. However, the treatment group that received transference interpretation had a slightly larger and statistically significant reduction in depressive symptoms from 12 weeks in treatment to one-year follow-up.

Whether the treatment effect on depression severity is consistent with the treatment effect on reducing sleep disturbance is to be examined in this paper. One may hypothesize that work on relational issues in the here and now may contribute to a more robust sense of self, less anxiety, and less ruminations about social relationships, which may contribute to better sleep quality [9]. 

This paper reports on data from the randomized controlled trial for adolescent depression, FEST-IT, where there was a high baseline prevalence of sleep disturbance. In extension of the IMPACT and FEST-IT trials, this paper aims at contributing to enhance our knowledge of how psychological treatment for depression may help reduce co-occurring sleep problems in youth. The aim is threefold. First, to describe the level and type of insomnia symptoms in a sample of adolescents diagnosed with MDD within the FEST-IT trial [15], measured by self-report questionnaires at baseline. Second, to explore how the sleep symptoms are related to symptoms of depression during the course of treatment and at one-year follow-up. Lastly, this paper will investigate the effect of transference work on sleep disturbance by examining the changes in sleep symptoms between the two treatment groups in FEST-IT.

## 2. Materials and Methods 

### 2.1. The First Experimental Study of Transference Work—In Teenagers (FEST-IT)

The adolescents in this study were taking part in a multi-center, randomized clinical trial on short term psychoanalytic psychotherapy (STPP) for adolescents with major depressive disorder. The First Experimental Study of Transference Work—In Teenagers (FEST-IT) study has a dismantling study design, where one specific treatment component was experimentally controlled [16], namely working explicitly with the therapist–patient relationship in the here and now of the session (transference work).

#### 2.1.1. Patient Material

Patients were recruited among adolescents with symptoms of depression who were referred to child and adolescent outpatient clinics in the South-Eastern Health Region, primarily in the capital Oslo and in the mixed urban and rural area of Vestfold in Norway. According to the ethical approval of FEST-IT, the participating adolescents were first informed about the study after fulfilling the formal criteria for MDD [16]. Written informed consent was then obtained from all patients. Exclusion criteria were comorbid psychosis, substance abuse, learning difficulties, and pervasive developmental disorders. A total of 70 adolescents aged 16–18 years diagnosed with a severe unipolar disorder according to Diagnostic and Statistical Manual of Mental Disorders, Fourth Edition (DSM-IV) [6] were included in the study. One patient withdrew the consent to participate. There were 57 girls and 12 boys, all attending classes in secondary school or high school. 

Comorbidity was frequent among the adolescents [16]. Table 1 shows relevant patient characteristics. Axis I conditions were assessed using the Mini International Neuropsychiatric Interview (M.I.N.I.) [17]. Axis II conditions were assessed using the Structured Interview for DSM-IV Personality (SIDP-IV) [18]. Pre-treatment characteristics on Axis I besides depression included anxiety disorders, primarily social phobia, and generalized anxiety disorder. Both proposed and recognized personality disorders (PDs) according to DSM-IV-TR [6] were measured among Axis II diagnosis. The mean GAF score [19] at the initial psychodynamic evaluation was 59.5 (SD  =  5.3 range 44.2–73.2). The mean GSI score (Global Severity Index from SCL-90-R) was 1.3 (SD = 0.5, range 0.5–2.7). The mean BDI score was 28.6 (SD = 9.1, range 10–58). Mean pre-treatment scores implied that the patient sample in this study on average had mild to moderate functional impairment and was moderately depressed. 

All medication was recorded at baseline, post-treatment, and at one-year follow-up. Only one patient reported taking antidepressant medication pre-treatment. One patient was taking sleeping medication at baseline. One patient reported taking antipsychotic medication throughout the study period. Post-treatment, one patient reported taking antidepressants; however, this was not the same patient as at baseline. Four patients reported taking sleeping medication at the end of the study [20].

#### 2.1.2. Treatment and Therapists

The short-term psychoanalytic psychotherapy (STPP) manual from the IMPACT study [21] was used as the treatment manual. STPP is a well-established treatment model in psychotherapy [22], and the manual combines aspects of STPP that primarily focus on techniques aimed at helping adolescents prevail developmental problems, and likewise accentuating the role of attachment theory, interpretation of unconscious conflicts, and the conception of internal working models. As stated in the study protocol [16], relational interventions were part of the treatment, for instance, through addressing the adolescent’s interpersonal transactions with others, or to interpret dynamic elements in the adolescent’s relationships with others. The youths were offered individual 45 min therapy sessions once a week, with a total treatment duration of 28 weeks. 

The adolescents were randomized into two treatment groups. The non-transference group (comparison group) received treatment with general psychodynamic techniques as described above. In the transference group, additional techniques for transference work were prescribed: (1) the therapist addressed transactions in the patient–therapist relationship, (2) the therapist encouraged the patients to explore thoughts and feelings about the therapy and the therapist, and (3) the therapist interpreted direct manifestations of transference and linked repetitive interpersonal patterns to the transactions between patients and therapist. These specific techniques were proscribed in the non-transference group. None of the treatment groups specifically considered how to address sleep disturbance as a routine intervention. 

Patients were assigned to one of twelve therapists. The therapists were eight psychiatrists and four clinical psychologists, six men and six women. All therapists were experienced and had at least two years of formal education in psychodynamic psychotherapy for adolescents. In addition, all therapists were trained in a one-year program in terms of specifically offering transference interventions in psychodynamic therapy and dynamic psychotherapy without transference work. 

#### 2.1.3. Evaluation

For a thorough and detailed description of the evaluation of the data in the FEST-IT study, please see the method section in the original paper [15]. 

### 2.2. Ethics

The Central Norway Regional Ethics Health Committee in Mid-Norway approved the study protocol (REK: 2020/21355 FEST-IT) 21.11.2011. Informed consent was obtained after the participants received information about the study. In accordance with Norwegian legislation, adolescents aged 16 years and older can make their own decisions regarding participating in health studies, and thus gave consent themselves. Participants in both treatment groups received well-established psychotherapy methods that are well described and frequently used. Trial registration: ClinicalTrials.gov Identifier: NCT01531101.

### 2.3. Measures in the Present Study

#### 2.3.1. Symptom Checklist 90 Revised (SCL-90-R) 

The SCL-90-R [23] is a 90 question self-report psychometric instrument designed to determine symptom severity and subjective psychopathology. It is widely used for research purposes, measuring outcome in psychiatric and psychological treatment. Psychometric properties have been described for adolescents [24], and the Norwegian version is well designed for assessing overall mental distress and changes in mental distress [25]. Patients in this study completed the SCL-90-R at pre-treatment, during treatment (after session 12), at post-treatment, and at one-year follow-up. In the present study, we use three of the specific questions related to sleep difficulties and use the mean value of these items as the primary outcome measure. The three questions are as follows: “Trouble falling asleep”, “Awakening in the early morning”, and “Sleep that is restless or disturbed”. These items tap into the key symptoms of insomnia during DSM-diagnosed major depressive disorder and constitute the number of sleep symptoms in this study.

#### 2.3.2. Beck Depression Inventory (BDI-II)

The BDI-II is a commonly used 21-item self-report instrument designed to measure specific symptoms of depression [26]. Psychometric properties have been described with the capacity to discriminate between depressed and non-depressed subjects, showing high reliability and structural validity in a variety of populations [27]. Patients in this study completed the BDI-II at pre-treatment, during treatment (after session 12), and at one-year follow up. The BDI-II was used as the secondary outcome measure in this study.

Only item 16 in the BDI-II contains a question about changes in sleep pattern in the last two weeks. Responses might be “sleeping more than usual” or “sleeping less than usual”. This item was excluded for the analyses in this study.

### 2.4. Statistics 

IBM SPSS (Statistical Package for the Social Sciences, Chicago, IL, USA) version 25 for Windows was used to run analyses. Variables were made by assembling three sleep-specific questions in SCL-90-R. The mean value of these questions was calculated for each patient at all measuring times. 

When assessing the relationship between continuous variables, Pearson’s r correlation coefficient was used to examine the correlation between self-reported depression depth in BDI-II and self-reported sleep problems in SCL-90-R. The cutoff for statistical significance was set at *p* < 0.05.

Differences in outcome across the two treatment groups with respect to sleep symptom/problems were assessed by the mixed models procedures in SPSS. This research question was evaluated by two different models, a linear model and a linear spline model. The linear model implies a straight line through all four measurement occasions. The linear spline model assumes that there are differences in change trajectories across time episodes. The choice of the “knot” was based on visual inspection of the change trajectories of the two groups (see Figure 1). This implies that treatment response was evaluated for two time episodes within the same model, i.e., (1) from baseline to post-treatment and (2) from post-treatment and one-year follow-up. 

Another momentum in model building was the choice of covariance structure. We proceeded along the same lines as in the first FEST-IT paper, comparing a marginal model for longitudinal data using an unstructured covariance pattern with a longitudinal random intercept model in which the covariance pattern is induced by including a random intercept [28]. Two fit indices were considered in evaluating the testing model fit: −2 log likelihood (−2LLH) and Akaike information criterion (AIC). Model parameters were estimated by maximum likelihood estimation. For the linear model with unstructured covariance, the −2LL was 545.3 and AIC = 571.3 (number of parameters = 13). The random intercept model (LL = 550.6, AIC = 560.6, number of parameters = 5). For the spline model with UN covariance, −2LL was 538.3 and AIC was 568.3 (number of parameters = 15). For the spline model with random intercept, −2LL was 544.7 and AIC was 558.7 (number of parameters = 7). Thus, the linear spline model with random intercept had the best model fit.

Time was coded as 0, 2, 3, and 8, representing four time points, i.e., pre-treatment, 12 weeks, 20 weeks, post-treatment, and one-year follow-up. Each integer represents approximately two months. 

#### Missing Data

At baseline, 68 (98%) participants had available and complete data sets for SCL-90-R, 58 (84%) participants had complete data sets at post-treatment, and 46 (67%) had complete data sets one-year follow-up. With regard to BDI, six patients had missing data at baseline. These data were imputed by linear regression using MADRS scores, describing in more detail the supplement of the original trial [15]. Eleven patients had missing BDI at post-treatment and 23 at one-year follow-up. Analyses were conducted of the frequency of the missing data across treatment groups. In a previous paper, we have shown that data are probably «missing totally at random» [15].

## 3. Results

### 3.1. Descriptive Statistics of Sleep Problems at Baseline 

On examination of subjective ratings of sleep problems in SCL-90-R at baseline, the most reported symptom was “difficulties falling asleep”—69% answered moderately to extremely, see Table 2 and Table 3 for details. The least commonly reported symptom was “Early morning awakening”. There were no statistical significant differences between treatment groups for frequency of baseline sleep problems, F(1,66) = 0.39, *p* = 0.53, n_p_^2^ < 0.01.

### 3.2. Correlation between Depression and Sleep

A Pearson correlation coefficient was computed to assess the relationship between sleep symptoms and depression depth over time for all patients. Depressive symptoms were assessed by the BDI-II self-report forms. The correlation between sleep symptoms and depressive symptoms at baseline was very small and non- significant r(56) = 0.122, *n* = 68, *p* = 0.321. There was a significant positive moderate correlation at session 20, r(35) = 0.497, *n* = 37, *p* = 0.002 and at the end of treatment, r(56) = 0.348, *n* = 58, *p* = 0.007. There was also a statistically significant moderate positive correlation between depression symptoms and sleep symptoms at follow-up, r(43) = 0.520, *n* = 45, *p* < 0.001. 

### 3.3. Change Trajectories for the Two Treatment Groups

Table 4 gives the results of the final longitudinal model comparing treatment outcome across the two treatment groups with respect to sleep symptoms. Table 5 shows the effect of “time1” was highly significant, whereas the interaction “time1 × treatment” was not significant. These findings indicate that both treatment groups had a significant change during treatment and that there was no significant change in rates across the two groups. The same applies for the second treatment episode (time2, from post-treatment to one-year follow-up), in which the interaction between time2 and treatment was not significant. Thus, as none of the interactions were significant, the change rates did not differ significantly across the two treatment groups for either time period. 

## 4. Discussion

The results of this study show that symptoms of sleep disturbance are highly prevalent among a sample of adolescents diagnosed with major depressive disorder. Using self-report forms measuring symptoms of insomnia, the most reported sleep symptom was initial insomnia. Contrary to expected, symptoms of insomnia did not correlate significantly with self-reported depressive symptoms at baseline. However, these correlations were moderate and statistically significant at the other three measurement occasions, with the largest correlation at one year-follow up. There were no statistically significant differences in recovery of sleep disturbance across the two treatment groups, i.e., the transference group and the non-transference group. Both treatment groups had a significant reduction in sleep problems during treatment, but not after treatment, i.e., from post-treatment to one-year follow-up.

In accordance with the results from the related IMPACT study, we found that the most frequently reported insomnia symptom on baseline was difficulties initiating sleep [1]. Measuring only insomnia symptoms, adolescents in the current study had on average a lower occurrence of sleep disturbance than adolescents in the IMPACT study. The overall level of insomnia symptoms in our patient material was, however, found to be approximately 15% higher than that of previously reported levels by Urrila et. al. and Liu et. al. [10,13]. The findings support that sleep disturbance is highly common in the presentation of MDD in adolescents, as seen in previous studies [29,30].

We hypothesized that self-reported depressive symptoms would be significantly correlated with the level of sleep symptoms during the whole study period. A possible explanation for the lack of association between self-reported sleep symptoms and depressive symptoms at baseline is that depression itself creates negative cognitive biases, leading to inaccurate reporting of symptoms and over-estimation of sleep problems [31]. However, if it is correct that patients with more severe depressive symptoms over-estimate their sleep problems, there should have been a stronger correlation at baseline. Thus, another hypothesis may lay within the etiological heterogeneity in MDD [32]. Young participants with a depression subtype characterized mainly by reactive features (occurring in response to a prior stressor) at baseline may be more likely to respond to psychotherapy than participants with endogenous depression [33]. Endogenous depressions are characterized by a greater degree of vegetative symptoms, including sleep disturbance. Perceived sleep quality may be distorted on baseline self-report forms owing to high levels of depressive symptoms. Changed perceptions of sleep after treatment for depression may contribute to more reliable reporting of sleep symptoms.

Exploratory analysis indicated no differences in recovery of sleep disturbance between the transference and non-transference group based on change rates. Adolescents in the present study were randomized to receive STPP for depression with or without transference interventions. We could not find evidence for the hypothesis that work on relational issues in the here and now contributes to less ruminations about social relationships, and thus better sleep quality. For the secondary outcome measures in the original FEST-IT trial, the transference group presented significantly better outcomes on BDI and MADRS, measuring depression severity, from 12 weeks in treatment to one-year follow-up [15]. An equal difference between treatment group is not seen in the current analyses when using solely the sleep items in SCL-90-R as an outcome measure. It is important to acknowledge that sleep problems naturally may improve over time [1].

The statistical non-significance between the treatment groups in the present study may be related to a small sample size. Another explanation may lay within the etiological heterogeneity in MDD, which has been suggested to lead to heterogeneity in treatment response [33].

With regard to the young individuals that experienced residual symptoms of depression and sleep difficulties at the end of the study period, we suggest that psychotherapy alone might be insufficient. Recent findings in treatment research show that adolescents with severe depression and endogenous features constitute a subgroup that are more likely to respond to pharmacotherapy than patients with less severe depression [34]. There is a need for more research on how to angle evidence-based treatment for these adolescents. 

### Strengths and Limitations

Several strengths and weaknesses should be acknowledged in this study. Liberal inclusion criteria in FEST-IT ensured that this sample is representative regarding the complex psychological challenges and comorbidity in outpatient clinics, and thus has relevance for clinicians [35]. Patients were not exclusively recruited for the study, but agreed upon participation after meeting the threshold for depression in outpatient clinics. Furthermore, gender differences known from Norwegian outpatient clinics reflect the gender imbalance (84% girls) in the data used for this study [36]. This might, however, also serve as a potentially confounding variable as studies on insomnia show a gender imbalance in insomnia suffering and MDD [37].

This study had several limitations. First, patients were categorized as having sleep disturbances based on three sleep items in SCL-90-R, which primarily screens for insomnia symptoms. Use of these items precludes considering other comorbid sleep disturbances associated with depression, such as hypersomnia. Furthermore, other medical conditions such as comorbid pain conditions [38] or obstructive sleep disorders [39], known to greatly interfere with sleep satisfaction, would have been overlooked with this method. It is, however, not uncommon in studies for insomnia to be assessed by few items [40]. 

As indicated previously, self-reported sleep problems may be subject to multiple biases such as common rater effect and measurement context effects [41]. There is evidence [40] that utilizing sleep diaries or objective measures such as polysomnography or actigraphy [42] provide more accurate information about sleep. These types of measures were not included in this study because assessment of sleep problems was not the primary aim of FEST-IT. However, authors like Mayers and coworkers [31] have argued that subjective perceptions of sleep in depression may be as important as objective measurements for sleep impairment owing to the lack of knowledge of subjective complaints in adolescent sleep research [43,44,45]. The experience of having a problem might not be captured only using objective sleep measures. 

Finally, the small sample size and the high frequency of missing data are limitations of this study. As pointed out earlier, we have outlined that missing data probably are “missing totally at random” and may not have had an impact on the overall results of this study. Further research should include a larger sample. 

## 5. Conclusions

There is a need to strengthen the knowledge of which interventions are effective for improving sleep disturbance in the context of depression in adolescence. This paper is among the first studies to explore the recovery of sleep disturbance following psychological treatment for depression in adolescents. Despite methodological limitations, we present preliminary data with clear treatment implications. Clinical data in this paper add to our understanding of how sleep disturbance is highly prevalent among a specific age group of adolescents diagnosed with MDD. The findings suggest that psychoanalytic psychotherapy for depression elicits a significant change in co-occurring sleep disturbance, regardless of whether or not the adolescent is receiving transference interventions. We suggest that psychotherapy alone might be insufficient for a subgroup of adolescents that experience both residual symptoms of depression and sleep difficulties at the end of the study period. Adolescents presenting with depression and sleeping difficulties in a clinical context should be broadly assessed for comorbidity. Future work with a range of sleep measures is required for early identification, correct assessment, and appropriate treatment of sleep disturbance.

## Figures and Tables

**Figure 1 ijerph-19-01790-f001:**
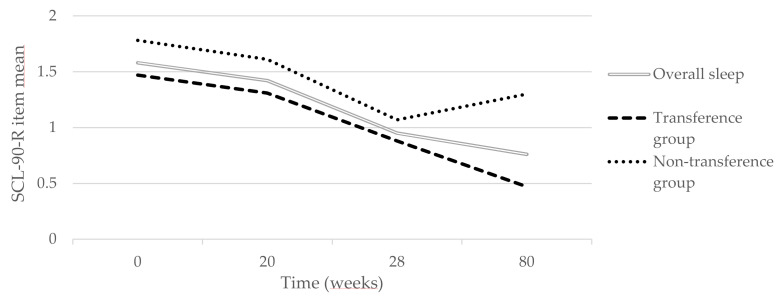
Changes in mean sleep quality (SCL-90-R) (%).

**Table 1 ijerph-19-01790-t001:** Pretreatment characteristics of the patients included in FEST-IT.

	Transference Work Group (*n* = 39)		Non-Transference Work Group (*n* = 30)	
	N	%	N	%
Female gender	33	84.6	24	80.0
Axis I diagnosis				
Depressive disorder	39	100.0	30	100.0
Prevalence of one or more comorbid diagnoses	18	46.2	16	53.3
Social phobia	7	17.9	3	10.0
Panic disorder	2	5.1	4	13.3
Agoraphobia	5	12.8	3	10.0
PTSD	1	2.6	1	3.3
	Mean (N = 39)	(SD)	Mean (N = 30)	(SD)
Age	17.3	0.7	17.3	0.7
BDI	28.5	9.5	28.8	8.3
GAF ^1^	59.6	5.5	59.2	5.0
MADRS *	20.8	4.5	24.0	6.1
GSI	1.2	0.4	1.6	0.6
Axis II diagnoses as measured with SIDP-IV (PD criteria)	13.5	9.0	12.4	7.8

^1^ Based on evaluator’s rating; * based on evaluator’s and therapist’s rating. Abbreviations: BDI, Beck depression inventory; GAF, global assessment of functioning; MADRS, Montgomery and Asberg Depression Rating Scale; GSI, Global Severity Index (SCL-90-R); PD, personality disorders; SIDP-IV, Structured Interview for DSM-IV Personality.

**Table 2 ijerph-19-01790-t002:** Frequency of SCL-90-R sleep difficulties at baseline (*n* = 69) (%).

	Trouble Falling Asleep	Restless or Disturbed Sleep	Early Morning Awakening
Not at all	17.6	31.3	58.8
A little bit	10.3	16.4	13.2
Moderately	23.5	19.4	13.2
Quite a bit	22.1	19.4	7.4
Extremely	26.1	13.4	7.4

**Table 3 ijerph-19-01790-t003:** Mean scores of SCL-90-R sleep difficulties at baseline across treatment groups (*n* = 68).

	Trouble Falling Asleep		Restless or Disturbed Sleep		Early Morning Awakening	
	Mean	(SD)	Mean	(SD)	Mean	(SD)
Transference work group (*n* = 39)	2.07	1.54	1.54	1.41	1.05	1.45
Non-transference work group (*n* = 29)	2.59	1.21	1.86	1.48	0.72	1.06

**Table 4 ijerph-19-01790-t004:** Frequency of SCL-90-R sleep difficulties at baseline, session 20, end of treatment, and follow-up (%).

Number of Sleep Symptoms	Baseline (0 Weeks) *n* = 68	Session 20 (20 Weeks) *n* = 37	End of Treatment (28 Weeks) *n* = 58	Follow-Up (80 Weeks) *n* = 46
0	5.9	16.2	20.7	37
>0–1	28.0	35.1	34.5	26
>1–2	35.3	29.7	32.8	22
>2–3	19.1	10.8	12.0	15
>3–4	7.4	8.1	0	0

**Table 5 ijerph-19-01790-t005:** Results from linear spline models.

Sleep Difficulties	Estimate	SE	CI (95%)	t-Value	F-Value	*p*
From baseline to post-treatment						
Time1 ^a^	−0.15	0.05	−0.26 to −0.05	−2.8	15.9	0.005
Time1 × treatment ^b^	−0.05	0.07	−0.20 to 0.09	−0.74	0.5	0.460
Posttreatment to one-year follow-up						
Time2 ^a^	−0.07	0.04	−0.14 to 0.006	−1.8	1.1	0.069
Time2 × treatment ^b^	0.08	0.06	−0.04 to 0.20	1.3	1.7	0.185

^a^ Change rate for the transference group, e.g., an estimate of −0.15 for time1 indicates that, for every second month, the transference group had an average reduction of 0.15 points of self-reported sleep symptoms. ^b^ Difference in change rates for the two treatment groups, e.g., an estimate of −0.05 indicates that the non-transference group had a reduction in sleep symptoms that was 0.05 stronger than the transference group.

## Data Availability

The data presented in this study are openly available in https://www.med.uio.no/klinmed/english/research/projects/fest-it/index.html (accessed on 1 February 2022).

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
