# Peer review of "Do Sleep Disturbances Improve Following Psychoanalytic Psychotherapy for Adolescent Depression?"

_ijerph, 2022, doi:10.3390/ijerph19031790_

Round 1

Reviewer 1 Report

The paper is interesting to read, however, there are a few changes/corrections that should be addressed prior to publication.

  1. In the paper (page 3, lines 125-126), the authors stated that “Axis I and II diagnosis were based on the Mini International Neuropsychiatric Interview (M.I.N.I.)”, this statement is not clear and ambiguous, authors should clarify the statement with some brief explanation.
  2. The Axis I and II diagnosis results are not labeled in Table 1. After that, the information for pretreatment characteristics of male patients are missing in Table 1. It’s recommended to include the above-mentioned information in the table.
  3. The data presented in Figure 1 is not clear, it would be better to label x- and y-axis of the graph?
  4. Page 9, line 333-334: “..sleep problems naturally may improve over time” -- provide a citation to support the statement.
  5. The conclusions are rather short. This could be improved by expanding the summary with some explanation on treatment implications in the conclusion.

Typographical comments:

  • Page 7, line 269: “session 20 end of treatment and follow-up (%)” -- missing comma after the word 20.
  • Page 7, The “Table 3. Frequency..” is wrongly labeled, it should be Table 4; Page 8, “Table 4. Results from linear...” should be Table 5.

Author Response

Dear peer reviewer,

First, we are extremely grateful for the helpful and highly constructive report. Thank you for taking the time to assess our manuscript.

You have suggested a few corrections that should be addressed prior to publication of the manuscript. We have addressed all the concerns that you have raised in your report. Please see the detailed list below.

  • Statements regarding axis I and axis II diagnosis have been clarified (page 3, lines 125-126) by dividing the original sentence in two.
  • Table 1 have been adjusted with an overhead to the axis I-diagnoses, as well as replacement of the label “PD criteria” to “Axis II diagnoses as measured with SIDP-IV (PD criteria).
  • Pretreatment characteristics of both male and female patients are already included in the table, as presented by the percentage of total participants.
  • As suggested, a citation to support the statement “sleep problems naturally may improve over time” has been included.
  • The conclusion has been expanded with some explanation on treatment implications.
  • We apologize that the data presented in Figure 1 was unclear. We have modified the figure and hope the labelling now is more explanatory.
  • Furthermore, typographical comments have been adjusted as sharply suggested by the reviewer – and the language has overall been revised.

Thea Schønning  

Reviewer 2 Report

This is a very well written manuscript describing a study where the authors  compared transference to non-transference psychotherapy and found that both forms of therapy helped alleviate sleep disturbances in adolescents with depression. The author should be applauded on their important work in helping adolescents with depression in this manner, as well as for bringing these findings to publication so others may benefit as well. One considerations I might offer, especially in the patients who did not improve in sleep symptoms, is to suggest that some of these patients may be experiencing problems with obstructive sleep disorder breathing in the form of mouth breathing, snoring, upper airway resistance, or obstructive sleep apnea. Disordered breathing is a common cause of depression and sleep-disturbance which could be treated by medical or dental sleep specialists.

Author Response

Dear peer reviewer,

First, we are extremely grateful for the supportive and highly constructive report. Thank you for taking the time to assess our manuscript.

You have provided a few considerations that should be addressed prior to publication of the manuscript. We have addressed all the concerns that you have raised in your report.

As suggested, the conclusions have been expanded with some explanation on treatment implications, especially regarding the adolescents that experience residual symptoms of sleep disturbance and depression after treatment. We are grateful for the consideration regarding adolescents that experience obstructive sleep disorders – and we have briefly included a sentence about this important subject (page 11, line 383) to raise concern.

Thea Schønning  

Reviewer 3 Report

The results of thisstudy show that symptoms of sleep disturbance are highly prevalent among a sample of adolescents diagnosed with Major Depressive Disorder. Using  self-report form s measuring symptoms of insomnia, the most reported sleep symptom was initial insomnia

There is a growing body of literature that recognizes the importance of sleep disturbance, and how it is related to adolescent depression

The bidirectional relationship between sleep disturbances and mood disturbances is known, but not always considered

There is a need to strengthen the knowledge of which interventions is effective for improving sleep disturbance in the context of depression in adolescence. This paper is  among the first studies to explore recovery of sleep disturbance in following psychological  treatment for depression in adolescents

My concerns:

Why did you have not considered an actigraphic study?

Adolescents often have an abnormal circadian rhytm that can be confused with insomnia.

Or Why did you have not used the Morning evening questionnaire?

You have to consider this issue in the limitation study.

The article addresses an important emerging issue

In the context of managing adolescent behavior, sleep disturbances must be considered absolutely
Adolescentanage of dangerous vulnerability

Author Response

Dear peer reviewer,

First, we are extremely grateful for the helpful and highly constructive report. Thank you for taking the time to assess our manuscript.

You have suggested a few corrections that should be addressed prior to publication of the manuscript. We have addressed all the concerns that you have raised in your report.

More specifically, some concerns were raised regarding the methodology of this study:

  • The clinical interview and self-report forms from which these data were gathered was unfortunately not designed with the assessment of sleep patterns and disorders as a primary goal. As a consequence, we have reported on secondary analyses and did not have the influence to include actigraphy or Morning evening questionnaire in the current study, although this would have been better for the aim of this study. This aspect has been included as limitations in the study, as cleverly suggested.
  • We agree with the reviewer that abnormal circadian rhythm might be confused with insomnia. In order to structure the current paper, we have decided not to discuss circadian rhythms further, as it constitutes its own, big study field.
  • Furthermore, the language has overall been revised.

Thea Schønning